# Effect of Coating Shell on High-Frequency Polarization Loss of Core-Shell Filler Dielectric Composites: An Alternating-Field Polarization Phase-Field Simulation of BN@SiO_2_/PTFE Composite

**DOI:** 10.3390/ma16155418

**Published:** 2023-08-02

**Authors:** Wenhao He, Yu Qi, Jie Shen, Xian Chen, Ming Fan, Jing Zhou, Wen Chen

**Affiliations:** State Key Laboratory of Advanced Technology for Materials Synthesis and Processing, Wuhan University of Technology, Wuhan 430070, China; hewenhao@whut.edu.cn (W.H.); 270506@whut.edu.cn (Y.Q.); 331249@whut.edu.cn (X.C.); fmky@whut.edu.cn (M.F.); zhoujing@whut.edu.cn (J.Z.)

**Keywords:** BN@SiO_2_/PTFE composite, polarization loss, dielectric mismatch, phase-field model, filler coating

## Abstract

Introducing a coating shell between the filler and matrix is an effective way to reduce the dielectric loss of the particle/matrix dielectric composites. It found that besides the improvement in interface compatibility, there may be some other effects of the coating shell, such as the elimination of the dielectric mismatch. However, the specific mechanism is still unclear due to the absence of an effective model for the quantitative analysis of the relationship between core–shell structure and dielectric loss, hindering the progress of the dielectric composite design. Here, a phase-field model for simulating high-frequency, alternating-field polarization is employed to study the relationship between high-frequency polarization loss and the coating shell in the silicon dioxide coating boron nitride polytetrafluoroethylene-based (BN@SiO_2_/PTFE) composite. The results show that the dielectric mismatch makes the high-frequency polarization loss spatially localized and periodically time-variant. The reduction of polarization loss depends on the polarization loss of SiO_2_. To reduce the high-frequency dielectric loss of the composite, the coating shell should not only eliminate the dielectric mismatch, but its dielectric loss must also be lower than that of the core filler. Furthermore, the model provided in this work has the potential to extend the quantitative calculation of non-intrinsic polarization loss and conduction loss.

## 1. Introduction

Boron nitride/polytetrafluoroethylene (BN/PTFE) composite attracts much attention in the application fields of high-frequency and high-speed signal transmission due to their outstanding dielectric and thermal properties [1,2,3]. To meet the developing requirement of advanced high-frequency substrates, high filler content is employed to tune the thermal conductivity, which unfortunately leads to an increase in high-frequency dielectric loss [4,5]. High dielectric loss in high filler content composite, which may originate from the dielectric mismatch between the filler and the matrix, is one of the current hot issues and has been discussed widely [6,7,8]. Recently, filler coating has been regarded as an effective way to eliminate dielectric mismatch [9,10,11,12]. Silicon dioxide (SiO_2_) has a lower dielectric constant than BN, and the SiO_2_ coating might influence the high-frequency dielectric loss of the BN/PTFE composite [13,14,15]. However, how the dielectric mismatch optimization influences the high-frequency dielectric loss is still unclear. The research on dielectric mismatch optimization stops at expounding that the local electric field distribution is regulated since the relationship between the core–shell structure and high-frequency dielectric loss is difficult to be analyzed quantitatively [16,17]. On the one hand, the high-frequency dielectric loss, affected by multiple factors, consists of multiple components, and the mechanism analysis of dielectric loss reduction should be finished by analyzing each component separately while influence factors are controlled; on the other hand, the relaxation polarization cannot be directly simulated by the Maxwell equations. Therefore, it is an emergency to develop a model with the capacity of simulating the relaxation polarization of BN@SiO_2_/PTFE composites under an alternating field.

In recent studies, based on the model of Semenovskaya [18], the phase-field model for the simulation of anisotropic dielectrics relaxation polarization has been developed [19,20,21,22], enabling the study of switching characteristics under pulsed electric fields [23], the simulation of variable temperature ferroelectric response [24], and cross-scale integration with the first-nature principle [25]. However, each component material in the BN@SiO_2_/PTFE composite is isotropic [26,27,28], which does not match the requirement of the previous models. Although Wang achieved the relaxation polarization phase-field simulation of the isotropic dielectric composite under the electrostatic field [29,30], there is still a lack of an alternating-field polarization phase-field model that can simulate the relaxation polarization of isotropic dielectric composite under the alternating field.

In this work, an alternating-field polarization phase-field model is set up and tested using experimental data. For the spherical BN@SiO_2_/PTFE composite, investigations into the impact of dielectric mismatch on high-frequency polarization loss and a quantitative analysis of the relationship between SiO_2_ coating shell and high-frequency polarization loss are conducted.

## 2. Simulation Methods

### 2.1. Alternating-Field Polarization Phase-Field Model

Based on the phase-field model of Wang et al. [29,30], the time-dependent Ginzburg–Landau equation is selected as the control equation:(1)$$\frac{\partial P(r,t)}{\partial t}=-L\frac{\delta F}{\delta P(r,t)}$$
where *t* represents the time, *r* represents the spatial position vector, *L* represents the dynamic coefficient, and *F* represents the total free energy function of the model.

In the equilibrium state, the Euler–Lagrange equation is satisfied. By combining the local electric field expression $E(r)=E_{ex}(r)+E_{d}(r)$ and the constitutive relationship $P\left(r\right)=\varepsilon_{0}\chi \left(r\right)E\left(r\right)$, the following can be obtained:(2)$$\frac{\delta F}{\delta P(r)}=\frac{\partial f}{\partial P(r)}-\nabla \cdot \frac{\partial f}{\partial \nabla P(r)}=\frac{P(r)}{\varepsilon_{0}\chi (r)}-E_{ex}(r)-E_{d}(r)=0$$
where $E_{ex}(r)$ represents the external electric field, $E_{d}(r)$ represents the depolarized electric field, and *f* represents the free energy density functional. Based on the variational principle, the total free energy functional *F* of the model is obtained as follows:(3)$$F=\int \frac{P^{2}(r)}{{2\varepsilon }_{0}\chi (r)}d^{3}r-\int E_{ex}{P(r)d}^{3}r-\int f_{d}d^{3}r$$
where $\varepsilon_{0}$ represents the vacuum dielectric constant, χ(r) represents the polarizability related to the spatial position, the relative permittivity related to the spatial position satisfies the relation ε(r)=χ(r)+1, and $f_{d}$ represents the potential energy density functional produced by the depolarized electric field.

At any time, the model always satisfies the Gauss law of dielectrics (Maxwell’s first equation), ∇·D(r)=0, and the divergence relation is obtained as follows:(4)$$\nabla \cdot E(r)=-\frac{\nabla \cdot P(r)}{\varepsilon_{0}}$$

By combining the expression into $\;E(r)=E_{ex}(r)+E_{d}(r)$, the expression of the instantaneous depolarization field $E_{d}(r)$ can be obtained using the spectral method as follows:(5)$$E_{d}\left(r\right)=-\frac{1}{\varepsilon_{0}}\int \frac{{{\{P}_{i}\}}_{k}q_{i}}{({2\pi )}^{3}q_{j}}e^{ik\cdot r}d^{3}k-\int \frac{{\left\{\nabla \cdot E_{ex}(r)\right\}}_{k}}{({2\pi )}^{3}q_{j}}e^{ik\cdot r}d^{3}k$$
where ***k*** represents the position vector of Fourier space, ***q*** = ***k/****k* represents the unit direction vector in Fourier space, and { }***_k_*** represents the forward Fourier transform. Based on the variational principle, the $f_{d}$ and $E_{d}(r)$ satisfy $E_{d}(r)=\frac{\partial f_{d}}{\partial P(r)}-\nabla \cdot \frac{\partial f_{d}}{\partial \nabla P(r)}$. The $f_{d}$ is calculated as follows:(6)$$f_{d}=-\frac{1}{{2\varepsilon }_{0}}\int \frac{q_{i}{{\{P}_{i}\}}_{k}^{2}}{({2\pi )}^{3}q_{j}}e^{Ik\cdot r}d^{3}k-\int \frac{{\left\{(\nabla \cdot E_{ex}\left(r\right))\cdot P_{i}\right\}}_{k}}{({2\pi )}^{3}q_{j}}e^{ik\cdot r}d^{3}k$$

The polarization response process can be obtained by solving the model with the display Euler method. To ensure the unity and reliability of the data, the study uniformly used a two-dimensional grid of *Nx* × *Ny*, where *Nx* = *Ny* = 1024. The combination of BN and PTFE or SiO_2_ was uniformly assumed to be in a well-bonded ideal state in the study. Additionally, <***E****(**r**)*> = ***E****^ext^* was always satisfied.

### 2.2. Model Parameter Setting and Model Verification

In high-frequency applications, it has been demonstrated that the external alternating field of the dielectric approximates a transverse electromagnetic wave, where the electric field direction is always perpendicular to the transmission direction [31,32]. Therefore, in this work, the applied electric field was assumed to be a plane wave propagating along the x-direction, and the electric field direction propagates along the y-direction, which is expressed as follows:(7)$$E_{ex}(r,t)=E_{0}e^{i(k\cdot r-wt)}$$
where ***k*** is the spatial unit vector satisfying |***k***| = 2π/*λ*, *λ* is the wavelength, *w* is the external field angular frequency, and ***E***_0_ is the external field amplitude satisfying ***E***_0_ = 10^4^ V/m

According to the definition of complex dielectric constant, the effective complex dielectric constant is obtained as follows:(8)$$\varepsilon^{eff}(r)=\frac{D(r)}{\varepsilon_{0}E(r)}=\varepsilon \prime (r)-i\varepsilon \prime \prime (r)$$

According to the relationship between tan*δ* and the complex dielectric constant, the local tan*δ* is obtained as follows:(9)$$tan\delta (r)=\frac{\varepsilon \prime \prime (r)}{\varepsilon \prime (r)}$$

The effective tan*δ* of the material can be calculated as follows:(10)$${tan\delta }^{eff}=<tan\delta (r)>$$

In the pure phase state, the material is close to the ideal state and the dielectric loss is dominated by polarization loss. The kinetic coefficient describes the polarization response rate, which decided the polarization loss calculation when the dielectric constant had been determined. The kinetic coefficient of PTFE *L*_p_ and the kinetic coefficient of BN *L*_b_ can be obtained by comparing the experimental tan*δ* with the calculated effective tan*δ*, respectively. The dielectric constant is set to 2.04 for PTFE and 4.5 for BN. Under the external electric field at 40 GHz, the real part of the effective complex dielectric constant and the effective tan*δ* of pure PTFE and pure BN are calculated as shown in Figure 1.

The computed value of the real part of the complex dielectric constant, which is generally in accord with the observed dielectric constant, is unaffected by the kinetic coefficient *L*, as shown in Figure 1a,c. According to recent experiments in our group, the measured tanδ of pure PTFE and pure BN at 40 GHz are 2 × 10^−4^ and 8 × 10^−4^, respectively, which was measured using an Agilent HP8722ET microwave network analyzer (HP8722ET, Agilent, @ 1–40 GHz) with a microstrip line method, according to ICP-TM-650 2.5.5. Comparing Figure 1b with Figure 1d, it is found that the value of effective tan*δ* gradually decreases as the kinetic coefficient rises. At the kinetic coefficients *L*_p_ of 3500 F/(m∙s) and *L*_b_ of 6000 F/(m∙s), respectively, the effective tan*δ* of pure PTFE and pure BN is close to the measured result. Therefore, the kinetic coefficient can be set using this finding. The 40 GHz is retained as the outfield frequency.

Then, in order to check the calculation process, the single-particle model was used. By filling PTFE with spherical BN particles with a particle size of 1 μm, the variation diagram of <***E***>/<***E***^ext^> with time was obtained (see Figure 2). As can be observed from Figure 2a, the free charge density changes within the minimal range, and the illustration shows that its value is always zero, which demonstrates that the Gauss law of dielectrics is always satisfied in the calculation process. Meanwhile, the results in Figure 2b show that the value of <***E***>/<***E***^ext^> is always close to 1, which demonstrates that the hypothetical conditions of this study are always satisfied, and the ideal state is always maintained in the system.

Finally, in order to check the calculation result, the rand distribution model was used (see Figure 3). Compared with the Bruggeman [28] and Lichteneker [33] theoretical models, the real part of the effective complex permittivity (see Figure 3a) and the effective tan*δ* calculations (see Figure 3b) are very similar to the Lichteneker theoretical calculation, which shows that the computational results are consistent with the effective medium theory, proving that the model built in this work simulates the polarization process in the ideal state.

## 3. Results and Discussion

### 3.1. Effect of Dielectric Mismatch on High-Frequency Polarization Loss

Dielectric mismatch and interparticle interactions have an impact on the local electric field distribution and the production of high-frequency polarization loss [34,35]. To avoid the effect of interparticle interaction, the single-particle model was employed to study the dielectric mismatch affecting polarization loss. A plot of the effective tan*δ* versus time was calculated and is shown in Figure 4. The formation of polarization loss soon achieves dynamic equilibrium, and the effective tan*δ* varies periodically with time after the application of the external field. The dielectric mismatch leads to a non-uniform distribution of the local electric field and the local polarization intensity [36,37], and the periodicity of the polarization loss is related to it. To support this claim, distributions of local area polarization intensity, local area electric field intensity, and local tan*δ* after various durations of external field application were obtained.

The substantial polarization loss of the filler particle itself is the reason for the filler particle’s much larger local tan*δ* at the beginning of the external field application, as illustrated in Figure 5a. The value of the local tan*δ* in the filler particle progressively drops with a change in the external field, while the local tan*δ* in the matrix gradually rises and exhibits an uneven distribution (see Figure 5b,c). The distribution of the local tan*δ* starts to change to the initial state when the external field changes through half a cycle (see Figure 5d), displaying a periodic variation; this periodicity echoes the periodicity of the external field. Within each cycle, the polarization loss of the substrate and the filler exhibits an opposite changing trend, and the inhomogeneous distribution of the local tan*δ* indicates the spatially localized nature of the polarization loss. The effect of dielectric mismatch is further illustrated with the help of Figure 5e–l.

As shown in Figure 5e–l, the dielectric mismatch results in an inhomogeneous distribution of local electric field and polarization intensity—both the local electric field strength and the local polarization strength decrease and then increase—which is in line with the trend of the applied electric field strength. The overall local polarization intensity drops to zero after the polarization intensity inside the filler reaches a particular level, and the dielectric-mismatch-induced inhomogeneous local electric field persists in the matrix for a considerable amount of time. Therefore, the local polarization intensity in the matrix region where the local electric field is concentrated degrades more slowly than it does in the filler and other regions. This causes the local polarization intensity in this region to have a more backward phase, which causes the polarization loss in this region to change in the opposite direction from that in the filler. This opposite variation trend results in the contribution of the polarization loss of the filler and the matrix to the overall polarization loss, showing periodic weighting alternation with the change of external field, leading to the spatial localization and time-varying periodicity of polarization loss. Overall, the dielectric mismatch causes the uneven distribution and periodicity of the polarization loss, allowing the polarization loss in the matrix to have a large proportion over a period of time. It is very important to eliminate the dielectric mismatch to reduce the loss of matrix polarization.

### 3.2. The Relationship between SiO_2_ Coating Shell and High-Frequency Polarization Loss

The dielectric mismatch and the inhomogeneous distribution of the local electric field within the substrate could be eliminated after the filler coating [34]. SiO_2_ has a dielectric constant of 3.9, which lies between the PTFE and BN dielectric constant, eliminating the dielectric mismatch between PTFE and BN by coating the BN filler. However, the characteristics of the cladding layer itself have often been overlooked in previous studies. The tan*δ* of SiO_2_ fluctuates, influenced by the preparation process and measuring environment, which might take values from 6 × 10^−4^ to 10 × 10^−4^ at 40 GHz. Therefore, the kinetic coefficients of SiO_2_ *L*_s_ are adjusted to meet the different tan*δ* for the deeper analysis of the connection between core–shell structure and polarization loss (see Figure 6a). The calculated effective tan*δ* of pure SiO_2_ is 10.13 × 10^−4^, 7.73 × 10^−4^, and 6.13 × 10^−4^ at kinetic coefficients *L*_s_ of 4000, 5000, and 6000 F/(m∙s), respectively, which match the value interval of tan*δ* of pure SiO_2_. To match the filler morphology of the coating experiment in our group, the thickness of the coating shell layer was chosen at 20 nm, and the Schematic diagram of SiO_2_-coated spherical BN particles is shown in Figure 6b. The BN is coated using SiO_2_ with three kinetic coefficients, and the plot of the effective tan*δ* with filler content is calculated by combining the random distribution model for different kinetic coefficients (see Figure 6c). According to Figure 6c, when filler content is increased after the SiO_2_ coating, the degree of polarization loss changes more than it did for the BN/PTFE composite. Additionally, the difference of SiO_2_ to BN affects the change in polarization loss trend. When the dynamic coefficient of SiO_2_ *L*_s_ is larger, that is, the effective tan*δ* of SiO_2_ is lower than that of BN, the core–shell structure will reduce the polarization loss of the BN/PTFE composites, and the decrease increases with the increase of filler content. When the dynamic coefficient of SiO_2_
*L*_s_ is smaller, the changing trend of polarization loss of the BN/PTFE composite is opposite to that of the previous after SiO_2_ coating. When the effective tan*δ* of SiO_2_ is close to that of BN, the core–shell structure eliminates the dielectric mismatch but does not reduce the polarization loss. In order to see the difference before and after SiO_2_ coating more intuitively, the effective tan*δ* distribution of BN@SiO_2_/PTFE composites with different *L*_s_ at different time points is calculated by selecting the content of 40 vol% (see Figure 7).

In the random distribution model, the inhomogeneous distribution of polarization loss is caused by dielectric mismatch, and the concentration of polarization loss in the matrix is also influenced by interparticle interaction (see Figure 7a–c). As shown in Figure 7d–f, the smaller *L*_s_ leads to higher polarization loss within the coating shell layer, which is the main reason for the increase in polarization loss after SiO_2_ coating. Comparing Figure 7a–c with Figure 7d–f, the local tan*δ* distribution is nearly the same before and after SiO_2_ coating, and the similar tan*δ* between coating shell and filler is the main reason for this phenomenon. Comparing Figure 7a–c with Figure 7j–l, the larger *L*_s_ leads to a lower polarization loss within the coating shell layer; the polarization loss within the matrix region near the interface is especially reduced, which can be found by comparing Figure 7c and Figure 7l. This phenomenon is the main reason for the lower polarization loss after coating. These results are consistent with the results in Figure 6c; they illustrate that reducing the high-frequency polarization loss of the composite requires the coating shell to not only eliminate the dielectric mismatch but also to have lower polarization loss than the core filler. Furthermore, the thickness of the coating shell does not change the effect of the physical property of SiO_2_; the results would still apply even if the thickness was changed.

## 4. Conclusions

In summary, a phase-field model for alternating-field polarization was developed to further explore the connection between core–shell structure and high-frequency polarization loss. Based on the measured dielectric property of pure PTFE and pure BN, the parameter setting of the model and the verification of the results was completed. The high-frequency polarization before and after SiO_2_ coating was simulated. The results show that the inhomogeneous distribution of the local electric field makes the polarization loss spatially localized, and the weights of the polarization loss of the matrix and the filler alternate periodically with the external field, and both characteristics are caused by the dielectric mismatch. Since the polarization loss in the matrix has a large proportion over a period of time, eliminating the dielectric mismatch to reduce the loss of matrix polarization is the key to reducing the whole polarization. The core–shell structure of the BN@SiO_2_/PTFE composite eliminates the dielectric mismatch between BN and PTFE, but the reduction of the polarization loss also depends on the tan*δ* of SiO_2_. To reduce the high-frequency polarization loss of the composite, the coating shell should not only eliminate the dielectric mismatch but also have lower polarization loss than the core filler. The model developed in this paper has the potential to extend the quantitative calculation of non-intrinsic polarization loss and conduction loss.

## Figures and Tables

**Figure 1 materials-16-05418-f001:**
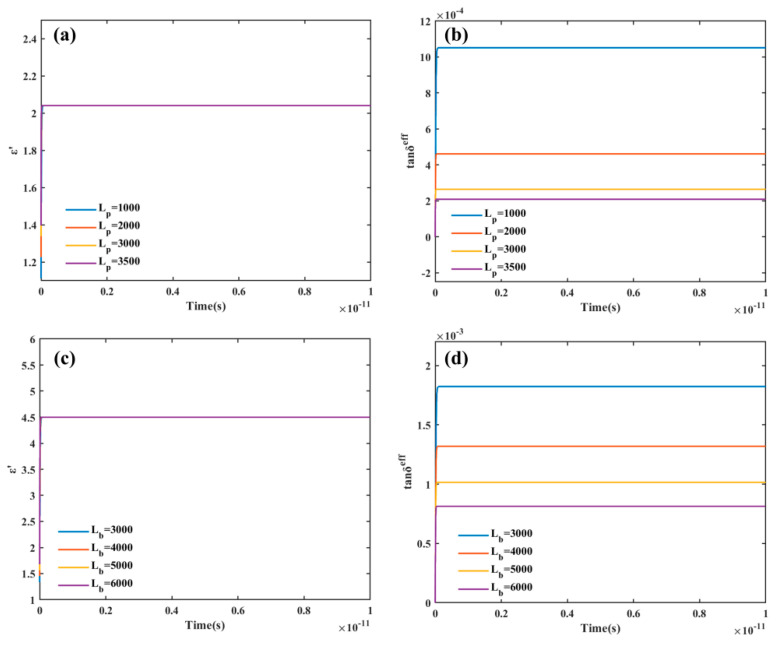
(**a**) The variation curve of the real part of the complex dielectric constant of PTFE with the different kinetic coefficients *L*_p_; (**b**) The variation curve of the effective tan*δ* of PTFE with the different kinetic coefficients *L*_p_; (**c**) The variation curve of the real part of the complex dielectric constant of the BN with the different kinetic coefficients *L*_b_; (**d**) The variation curve of the effective tan*δ* of BN with the different kinetic coefficients *L*_b_.

**Figure 2 materials-16-05418-f002:**
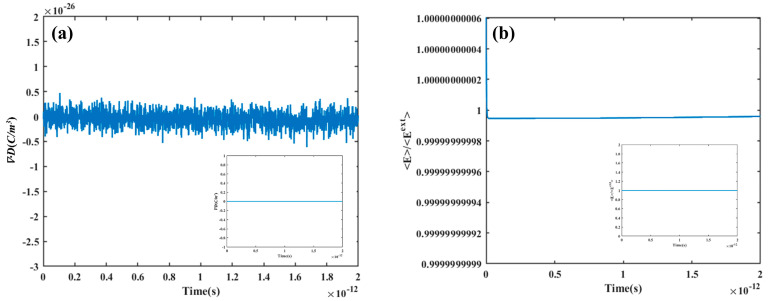
(**a**) The variation curve of ∇·D with time at single-particle model; (**b**) The variation curve of <***E***>/<***E***^ext^> with time at single-particle model.

**Figure 3 materials-16-05418-f003:**
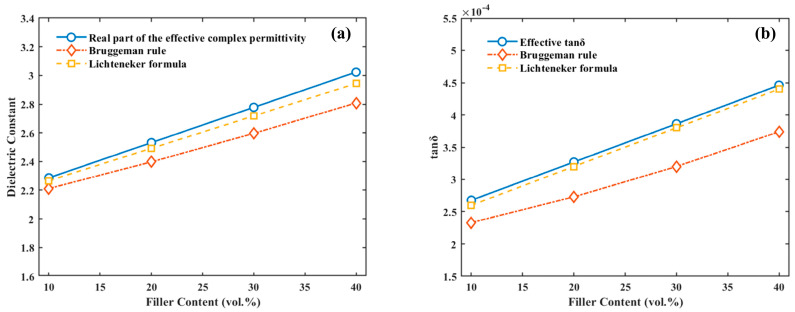
(**a**) Comparison of the real part of the effective complex permittivity with the Bruggeman [28] and Lichteneker [33] theoretical models; (**b**) comparison of the effective tan*δ* with the Bruggeman [28] and Lichteneker [33] theoretical models.

**Figure 4 materials-16-05418-f004:**
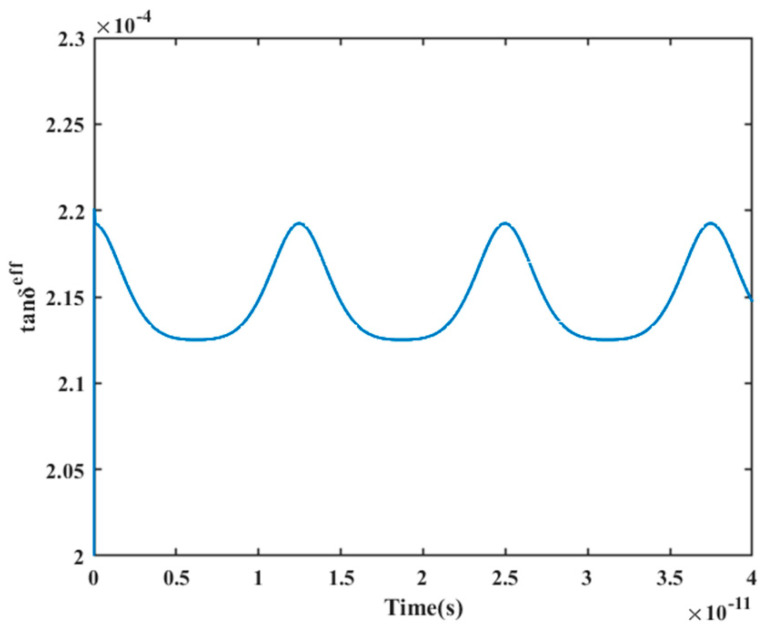
The variation curve of effective tan*δ* with time at the single-particle model.

**Figure 5 materials-16-05418-f005:**
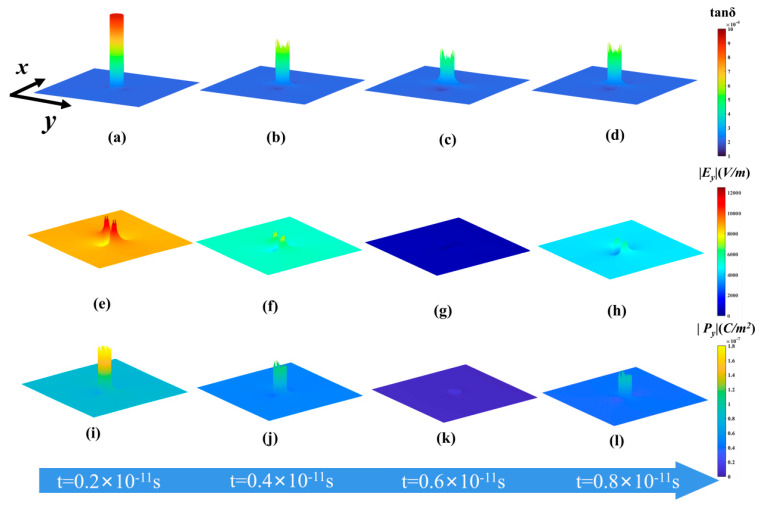
(**a**–**d**) Distribution variation of local tan*δ* with time in the single-particle model; (**e**–**h**) distribution variation of local electric field intensity with time in the single-particle model; (**i**–**l**) distribution variation of local polarization intensity with time in the single-particle model.

**Figure 6 materials-16-05418-f006:**
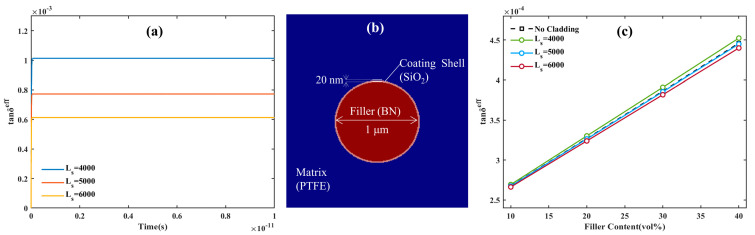
(**a**) The variation curve of the effective tan*δ* of pure SiO_2_ with time at different kinetic coefficient *L*_s_; (**b**) Schematic diagram of SiO_2_-coated spherical BN particles; (**c**) the variation curve of the effective tan*δ* with filler content at different kinetic coefficient *L*_s._

**Figure 7 materials-16-05418-f007:**
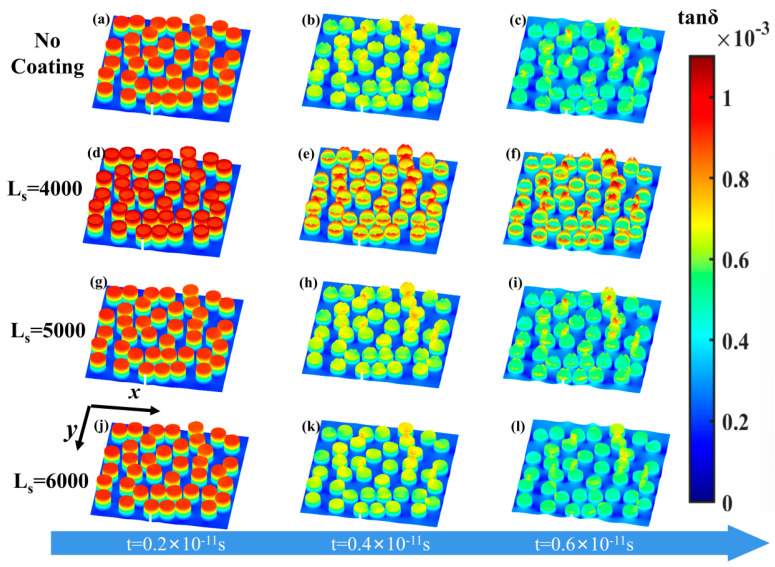
(**a**–**c**) Distribution variation of local tan*δ* of BN/PTFE composites with time in the random distribution model at 40 vol.%; (**d**–**l**) distribution variation of local tan*δ* of BN@SiO_2_/PTFE composites with time at different *L*_s_ in the random distribution model at 40 vol.%.

## Data Availability

The data presented in this study are available on request from the corresponding author.

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
