# Peer review of "Effect of Coating Shell on High-Frequency Polarization Loss of Core-Shell Filler Dielectric Composites: An Alternating-Field Polarization Phase-Field Simulation of BN@SiO2/PTFE Composite"

_materials, 2023, doi:10.3390/ma16155418_

Round 1

Reviewer 1 Report

 The work presented in this article is interesting and can be accepted for publication after some minor revision.

1. The authors should clearly write the name of the component materials once a time in the manuscript before using abbreviations to attract more attention and increase the reading of the study. ( ie., BN, PTFE )

2. Additionally, with a given Figure, the Schematic illustration of coating the particle/matrix dielectric composites can lead to a visual clarification of the study.

3. Why was the coating shell layer's thickness chosen at 20 nm? In similar studies given in the literature, the effects of different thicknesses on the results are evaluated. Are the results of this study valid for different thicknesses? Or are they variable depending on the model or material used? This clarification is significant for experimental studies and potential applications.

4.  In Section 3, a brief Comparison of the results with other simulation and experimental studies needs to be given to underline the novelty of the work.

5. In Section 3, how "the coating shell should not only eliminate the dielectric mismatch but also have lower dielectric loss than the core filler." is need to be clarified. This result must be explained in terms of physical phenomena.

Reviewer 2 Report

Athors of the study reported a phase-field model for simulating high-frequency alternating-field polarization in BN@SiO2/PTFE composite. Their study is of significancy for the material science. However, minor changes should be applied as follows:

1)   There is no description of the abbreviations at their first useage in the abstract and main text of the manuscript. For instance, the introduction is started by the phrase “BN/PTFE composite attract much…” without introducing abbreviation of the composite. Similar comment for the abstract.

2)      The quality of the figures should be improved. It looks lower than 300 DPI.

3)      The exact applications of the proposed composite need to be discussed in more details for the broad audience.

Reviewer 3 Report

This paper is to investigate the local polarization potential of SiO2 coated BN filler-PTFE composite using ac-field phase-field model. This is interesting, but its effectiveness should be verified with the many  experimental data. If possible, the more experimental data should be included and compared with the calculated ones. Moreover, the experimental procedure is not described anyplace, although the experimental data are presented in the text. The coating of SiO2 on the BN would weaken the local electric field, which reduce the loss. Could you tell the details of the mechanism on how to reduce the field. (by eliminating the potential point defects? or reducing surface electric potential?) And below concerns should be addressed before publication.

1.     In the equation (9), pls describes the difference between tan d and the effective tan d.

2.     In the page 3, its was written as “by comparing the measured tan d with the calculated tan d..”. Pls, describes how to measure the tan d under external electric field at 40 GHz. You have used the measurement of S-parameter at such a high frequency with a special attachment?

3.     The size and shape of BN fillers would affect the dielectric constant and loss. So, in this paper it is very curious about how to define the BN fillers. (filler as defined as sphere with a diameter of 1 micrometer?)

Round 2

Reviewer 3 Report

This paper was well revised, according to the comments of reviewer. Therefore, it can be recommended to publish in the Materials.